# Platelet-Rich Plasma for the Treatment of Degenerative Lumbosacral Stenosis: A Study with Retired Working Dogs

**DOI:** 10.3390/ani11102965

**Published:** 2021-10-14

**Authors:** Ángel María Hernández-Guerra, José María Carrillo, Joaquín Jesús Sopena, José Manuel Vilar, Pau Peláez, Belén Cuervo, Angelo Santana, Mónica Rubio

**Affiliations:** 1Departamento de Medicina y Cirugía Animal, Facultad de Veterinaria, Universidad Cardenal Herrera-CEU, CEU Universities, 46115 Valencia, Spain; angelhdez@uchceu.es (Á.M.H.-G.); jcarrill@uchceu.es (J.M.C.); jsopena@uchceu.es (J.J.S.); pau.pelaez@uchceu.es (P.P.); belen.cuervo@uchceu.es (B.C.); mrubio@uchceu.es (M.R.); 2Department of Animal Pathology, Instituto Universitario de Investigaciones Biomédicas y Universitarias, Universidad de Las Palmas de Gran Canaria, Trasmontaña S/N, 35416 Arucas, Spain; 3Departamento de Matemáticas, Universidad de Las Palmas de Gran Canaria, 35018 Las Palmas, Spain; Angelo.santana@ulpgc.es

**Keywords:** cauda equina, lumbosacral estenosis, platelet rich-plasma, PRP, epidural

## Abstract

**Simple Summary:**

The incidence of degenerative lumbosacral stenosis in dogs has been increasing in the last few years, mainly due to longer life expectancy. Given that degenerative lumbosacral stenosis is due to multifactorial causes, there is no consensus regarding the best treatment strategy. The aim of this study was to subjectively and objectively assess the efficacy of a series of epidural infiltrations of a platelet-rich plasma derivative. Fourteen retired working dogs with this syndrome were infiltrated, and clinical and force platform data were recorded. The results showed a significant clinical and gait improvement during the 90-day period. This treatment strategy seems to be a good alternative to other medical treatments, given that the autologous product is effective and considered innocuous.

**Abstract:**

Traditionally, canine degenerative lumbosacral stenosis (DLS) has been defined as a multifactorial syndrome characterized by lumbosacral pain triggered by the compression of the nerve rootlets of the cauda equina. There is still no consensus on the treatment of this condition, probably because there are a plethora of possible causes. In addition to compression, inflammation is a very important factor in the physiopathology of the disorder. Platelet-rich plasma (PRP) consists of an increased concentration of autologous platelets suspended in a small amount of plasma. Platelets are a source of several growth factors. Growth factors were shown to help in wound healing and biological processes, such as chemotaxis, neovascularization and synthesis of extracellular matrix, and growth factors were used to improve soft tissue healing and bone regeneration. PRP also facilitates the restoration of the structural integrity of the affected anatomy. Fourteen dogs diagnosed with DLS were treated with three epidural injections of PRP on days 0, 15 and 45. All dogs showed clinical improvement 3 months after the initial treatment. Gait was also objectively assessed by means of the use of force platform analysis before and after treatment, showing significant improvement. The results show that PRP may provide a good alternative to other nonsurgical treatments, such as prednisolone epidural injection.

## 1. Introduction

Degenerative lumbosacral stenosis (DLS) is an acquired multifactorial condition in which compression of the cauda equina nerve roots occurs as a result of alterations in the surrounding tissues that cause progressive stenosis of the vertebral canal and/or the intervertebral foramina. This compression leads to pain and, in the most affected patients, to neurological deficits [1]. There are a plethora of causes of DLS: spondylarthrosis, disc disease (mainly type II), interarcuate ligament hypertrophy and/or facet joint hypertrophy, varying degrees of misalignment of the vertebral body and congenital vertebral malformations (e.g., transient vertebra, fused vertebra), which invariably lead to an osteoarthritic process [2]. Probably due to the numerous causes and degrees of involvement described for the DLS, there is no consensus regarding its treatment, given that several different types of management were described, including conservative, surgical and epidural steroid injection (EI). Although it does not have a standardized protocol, conservative management is based on restricted exercise and oral medications, such as anti-inflammatories, analgesics and oral corticosteroids. For conservative management, reports describe success in relieving clinical signs in about 50% of dogs [3]. For surgical management, improvement was reported between 67 and 97% of cases; however, due to their multifactorial nature, different techniques were described, making them difficult to compare [4,5]. The third management option, EI of methylprednisolone, has only been described twice [6,7]. Given that there was a relapse of clinical signs at two months in 77% of the treated dogs, the main conclusion of these studies is that more than one EI is required for long-lasting results in a significant percentage of dogs. Apart from these three treatments, other authors reported, although still not sufficiently proven, intradiscal injections and physical rehabilitation [1,8].

EI is, in general, considered a safe procedure in dogs [9,10]. The aim of the EI is to deposit anti-inflammatory drugs directly onto the inflamed area, causing less systemic effects and higher local dosages [11]. Anti-inflammatory EI would address the fact that the pathophysiology of the lumbosacral is inflammatory rather than only compressive and mechanical [6,12]. These advantages, together with their rapid effects, make it an easy and relatively cost-effective management of DLS [7,13].

Platelet-rich plasma (PRP) is an autologous biological product derived from the patient’s blood, in which a plasma fraction with a higher platelet concentration than circulating blood is obtained after a centrifugation process. The basis of the procedure is that higher platelet concentrations release significant amounts of growth factors [14,15]. PRP and its derivatives are gaining interest in regenerative medicine due to its potential to stimulate and accelerate tissue healing and its ability to help in the regeneration and restoration of damaged tissues and cells [16].

In veterinary medicine, PRPs are mainly used in the treatment of musculoskeletal, ophthalmological and orthopedic pathologies [16,17,18,19].

Given that DLS leads to pain and/or discomfort, it directly causes a functional limitation, specifically with gait. Force platform (FP) analysis is considered the “gold standard” of objective gait assessment and has been widely used to detect lameness in osteoarthritic dogs, as well as to assess the objective efficacy of different therapeutical strategies. The most frequently used parameter is the Peak Vertical Force (PVF). When dogs of different conformations are used in studies, an accepted method to standardize the results is the use of symmetry indices [20].

For this study, we hypothesized that epidural PRP injections could be an alternative method of treating DLS, yielding comparable results to that of the methylprednisolone acetate injection previously described [9,10,11].

The purpose of this prospective study was to assess the efficacy of EI of a PRP derivate in dogs diagnosed with DLS by means of clinical and kinetic parameters. 

## 2. Materials and Methods

Fourteen working dogs were treated by means of an EI of PRP. The dogs were enrolled over a period of two years, between September 2018 and May 2020, at the Hospital Clínico Veterinario of the CEU—Universidad Cardenal Herrera. These patients had been diagnosed with DLS based on a complete physical, neurological examination and complementary diagnostic tests, consisting of a haemogram, biochemistry, a urinalysis, x-rays with ventrodorsal and lateral views and a CT scan. All dogs were treated medically for at least one month with a combination of rest, AINEs and gabapentin with unsatisfactory results; however, the dogs did not receive any additional treatment for one month prior to the study.

The study protocol was approved by the animal ethics committee of the government of the Valencia region (Spain). The approval code was 2019/VSC/PEA/0045. The dog owners provided written consent to participate in the study.

Dogs were considered to suffer DLS based on clinical assessment and imaging confirmation. 

### 2.1. Clinical Assessment 

Animals should show at least two of the following clinical signs: visual, subjective detection of paresis/weakness of the pelvic limbs; lumbosacral pain/hyperesthesia of the LS junction (the dog vocalizes or resists in response to dorsal digital pressure on LS junction when lifting the tail and/or when extending hips); urinary incontinence; hind leg withdrawal reflex deficiency; and proprioceptive deficit. Clinical examinations were assessed by the same researcher (AH).

### 2.2. Imaging Confirmation (X-ray and CT Scan) 

Imaging confirmation was obtained based on the presence of at least one condition considered a component of DLS (Figure 1). These conditions were previously published [1] (Table 1). 

Dogs with orthopedic conditions of the hind limbs, severe hind leg proprioceptive deficit or signs of any other neurological abnormalities unrelated to the cauda equina were excluded from the study. Signs of any chronic medullary conditions (i.e., degenerative myelopathy and disk protrusion further cranial) were especially taken into account. Dogs with severe paresis due to cauda equine were also excluded from the study, given that they are considered candidates for surgical resolution.

The clinical assessment detailed in Section 2.1 was used in every subsequent reexamination. 

EI protocol consisted of three epidural infiltrations: day (D) 0, the second at D15, and the third at D45. The days for clinical and FP assessments are shown in Figure 2. 

EIs were performed with a solution of PRP. How the PRP was obtained was described previously in [21] but briefly: the PRP obtention is based on the PRGF-*Endoret*® system (BTI Biotechnology Institute, Vitoria, Spain), which is a type of PRP with approximately 1.5 times the concentration of erythrocytes in whole blood and absence of leukocytes; after the blood collection in tubes with citrate, just one centrifugation at 460 g during 8 min was performed. Then, red blood cells were obtained at the bottom of the tube, just above white cells, and finally, two fractions of plasma; the upper 60% corresponds to the fraction poor in GFs, and below this, the remaining 40% corresponds to PRGF. Dogs were previously sedated with a combination of dexmedetomidine 300 mcg/m^2^ IM (Dexdomitor®, zoetis, Madrid, Spain) and butorphanol 0.4 mg/kg IM (Torphadine®, Produlab Pharma, Raamsdonksveer, The Netherlands). An epidurogram with 0.5 to 1 mL of contrast medium was used to confirm the correct location of the spinal needle. PRP was then infiltrated.

### 2.3. FP Assessment 

Functional gait analysis was performed using a single platform mounted in the center of, and level with, a 7m runway covered by a rubber mat. Dogs were leash guided at walk over the FP by the same handler. In order to collect data from dogs at the same “dynamic status”, velocity was progressively augmented to reach a maximum where dogs were still walking [22]. Once appropriate velocity was reached, three valid trials at a sampling frequency of 250 Hz were obtained for each dog. A trial was considered valid when the limb fully contacted the FP and when the dog walked next to the handler without pulling on the leash. The platform was interfaced with a dedicated computer using DataStudio® (Pasco, Roseville, CA, USA), which is software specifically designed for the acquisition, numerical conversion and storage of data.

Once PVF data were obtained, the mean value was used to set a symmetry index(SI) between contralateral hindlimbs using the following formula: SI% = 200 × (SL−LL)/(SL + LL), where SL is the PVF (in Newton) of the limb which showed a higher value, and LL is the PVF of the limb which showed a lower value. In this formula, 0% should represent perfect symmetry [20].

## 3. Results

Males represented 71% of dogs (10/14). The mean age was of 10 years (range 7–12), and the mean weight was 29.12 kg (range 2536). Dog breeds consisted of Labrador retrievers (4), Rottweilers (2), German shepherds (3) and mixed (5). All were retired working dogs, which fell under the following tasks: guard/obedience (5), guide (4), agility (3) and police (2).

### 3.1. Clinical Parameters 

Clinical signs found at D0, D15, D45 and D90 are summarized in the following table (Table 2). 

### 3.2. FP Analysis 

The mean values of PVF obtained from the LL and CL of each dog before and after treatment are shown graphically (Figure 3).

As can be seen, values from LL (lower dot) and CL (higher dot) before (red) are closer (more similar) after treatment (blue), meaning the gait is more symmetric. 

The SI obtained from PVF values of all dogs before and after treatment and differences among them are shown in the following table (Table 3).

## 4. Discussion

In this research study, the clinical improvement of dogs with DLS treated with an EI of PRP was subjectively and objectively proven. Like most previous reports, most dogs were male (10/14), over 25 kg and older than six years of age. These dogs presented paresis, or weakness of the pelvic limbs, and pain on lumbosacral palpation and/or when lifting their tails. All these findings are consistent with the results in other studies [1,6]. 

Our results coincide with those obtained with traditional treatments: conservative management, based on a change in lifestyle, including weight loss and restricted exercise, and medications, anti-inflammatories, analgesics and oral corticosteroids. In these cases, previous reports showed success in relieving clinical signs in about 50% of dogs [3,23]. The main advantage of this treatment is that autologous PRP therapy is considered practically harmless [24].

The pathogenesis of DLS is multifactorial and still not fully understood. The main factors that contribute to it include compression through disc protrusion/extrusion, ligament hypertrophy and changes due to degenerative joint disease, dynamic compression and others [1]. In particular, low-back pain, the major clinical sign of DLS, also seems to have a multifactorial origin. Nerve compression; damage to adjacent soft tissue structures, such as the annulus fibrosus; ligaments and synovial fluid; and inflammation also seem to account for the pain [25]. Prostaglandin E2 (PGE2), a pain mediator, is significantly increased in degenerated discs, even at early stages. For that reason, it would make sense to use anti-inflammatory drugs, such as PRP, close to the lesion to treat the pain by controlling inflammation [1]. 

Compared with other EI of anti-inflammatory drugs, the results of this study are even better than when compared with EI of methylprednisolone [6]. Even some neurological deficits, such as proprioceptive deficits and withdrawal reflexes, disappeared after the third injection, at least during the checking period. However, neither of the two dogs with urinary incontinence improved their condition. Interestingly, one dog improved its urinary incontinence after the first EI, only to relapse after the second injection. In this respect, some authors consider urinary incontinence as a bad prognostic sign [26,27].

Compared with studies of dorsal decompressive surgery in dogs [2,28], our treatment results seem better. However, clinical signs were more severe in the surgical cases than in this study, so a true comparison is difficult. Although there is insufficient data, it was suggested that surgical treatment may provide longer relief of clinical signs and would therefore be more suitable for younger working dogs [1]. On the other hand, administering corticosteroids would be more effective in helping to lengthen the career of dogs close to retirement. In average veterinary practice, most patients suffering from DLS are not working dogs but rather middle age to old animals; therefore, EI seems to fit perfectly as a treatment for the average canine profile suffering from DLS.

To achieve our clinical goals, PRPs must contain at least one million platelets per microliter and have a platelet concentration of 1.5 times that of whole blood [16]. The use of PRP and/or growth factors as a treatment for degenerative intervertebral disc disease in human medicine was also described, where the results of these treatments were mostly favorable with little or no adverse effects [29]. It has also revealed an anti-inflammatory effect, downgrading the expression of pro-inflammatory cytokines, such as tumor necrosis factor alpha (TNF-α), interleukin 6 (IL-6) and interleukin 1 beta (IL-1β). Moreover, the injection of PRGF significantly reduced levels of TNF-α and IL-1β gene expression and secretion in vitro [30]. This anti-inflammatory effect may be, at least partially, what lies behind its beneficial effects in the treatment of canine DLS.

In a human study comparing intra-articular injections of methylprednisolone and autologous leukocyte-free PRP, the results clearly favored the use of PRP. The differences were small during the first 4 weeks, but 3 months after the treatment, the PRP group had significantly more improvement [31]. This outperformance in the long term compared to locally injected steroids was also noted in another human study in patients with facet arthropathy [32].

It was shown that PRP derivatives applied in a confined space (i.e., joint cavity, vertebral canal) and in solid (activated) form have results that are more consistent and specific, especially if repeated injections are administered [33]. 

Among the different protocols of PRP preparation, leukocyte-poor/absent PRP derivatives seem to be the most effective, avoiding the pro-inflammatory effects of leukocytes; the concentration of stored platelets and growth factors after eliminating the erythrocytes and leukocytes will allow a faster and more efficient wound and tissue healing [34]. For that reason, we used PRGF-Endoret® technology.

In order to objectively compare the different subjects’ values, it was necessary that the animals move at equal relative (normalized) velocities. However, if dogs were walked at the same (given) velocity, smaller dogs needed to travel at a relatively higher speed than larger dogs [35] or change their gait pattern from walk to trot; for this reason, the appropriate speed should be when different dogs move at the same “dynamic status”. Given that a dog can walk at a wide range of velocities, even at the same gait, data were obtained from all dogs when they reached their maximum velocity at walk [22].

In this study, we used animals of different weights, which could become an obstacle to obtain reliable results through FP analysis; however, the normalization of these parameters using a SI to compare results between contralateral limbs in all the dogs should, according to the theory of *dynamic similarity*, enable the comparison of results between dogs of different sizes [36,37]. A SI of about 3% was set as a cutting point to discriminate between sound and lame dogs, in such a way that a dog with a SI above this number is already considered lame [38,39].

With these premises and based on our results, almost all the dogs significantly improved (reduced) their SI. Moreover, four dogs can be considered “sound” after treatment. However, given that the cutpoint of 3% SI can only be considered approximatively, the total dogs with a SI under 5% is raised to 10. This objectively could be interpreted as the number of animals that become sound or with a slight lameness after treatment.

## 5. Conclusions

EI of PGRF-Endoret® in working dogs with DLS with no or mild neurological deficits has proven to be subjectively and objectively effective; thus, this autologous, absent of side effects PRP derivative could be considered an additional therapeutical strategy in dogs suffering from this condition.

## Figures and Tables

**Figure 1 animals-11-02965-f001:**
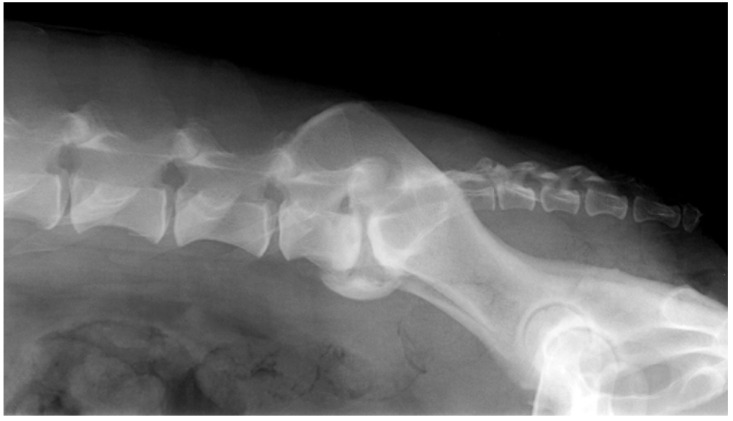
Laterolateral X-Ray image of dog # 4, neutral position. Severe spondyloarthritis, narrow intervertebral disc space and thickened and sclerotic endplates can be seen.

**Figure 2 animals-11-02965-f002:**
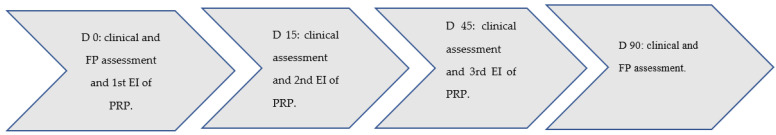
EI, clinical assessment and FP analysis workflow.

**Figure 3 animals-11-02965-f003:**
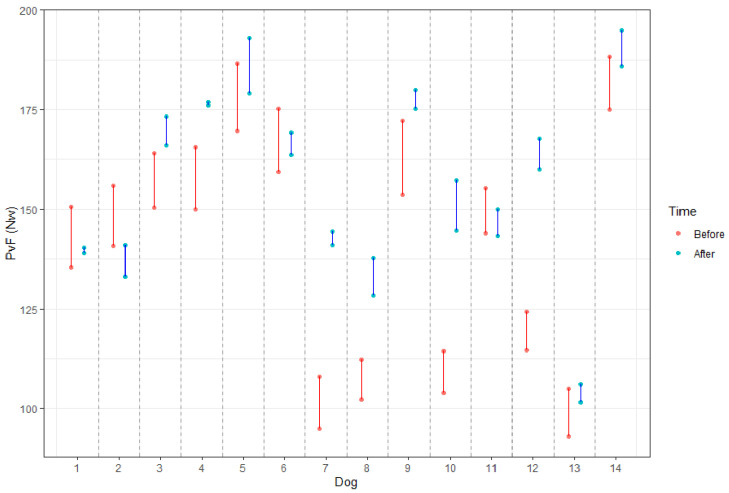
PVF obtained from LL and CL of each dog before (red) and after (blue) the treatment. The dog’s id number was randomly assigned.

**Table 1 animals-11-02965-t001:** Pathology recognized as a component of degenerative lumbosacral stenosis in dogs.

Hypertrophy of the ligaments stabilizing the LS junction (dorsal longitudinal ligament ventrally and *ligamentum flavum* dorsally).
Degeneration of the lumbosacral disc, with protrusion of the disc annulus.
Degenerative joint disease of the articular processes, with modification of the shape of the articular surface, periarticular new bone formation and hypertrophy of the joint capsule.
Lateral spondylosis deformans at the lumbosacral junction and sacroiliac joint, which can impinge into the exit zone of the L7-S1 intervertebral foramen and compress the L7 intervertebral neurovascular bundle.
Dynamic compression of the cauda equina caused by the ventral displacement of the sacrum in relation to L7 (step lesion = retrolisthesis).
Dynamic narrowing of the L7-S1 lateral intervertebral foramen during extension of the lumbosacral joint (telescoping).
Congenital stenosis of the vertebral canal at the LS junction.
Transitional vertebral anomaly.
Osteochondrosis-like lesion of L7 or S1.

**Table 2 animals-11-02965-t002:** Clinical signs before (D0) and after (D15, D45, D90) the treatment cycle.

Item	D0	D15	D45	D90
Paresis/and weakness of the pelvic limbs	12	2	2	0
Lumbosacral pain /Hyperesthesia				
Dorsal digital pressure	14	12	2	2
Lifting the tail	6	6	4	2
Extending hips	2	2	0	0
Urinary incontinence	4	1	2	2
Hind leg withdrawal reflex	8	8	4	0
Proprioceptive deficit	6	6	4	0

No post-injection side effects were reported.

**Table 3 animals-11-02965-t003:** This table includes the pathology, the SI (symmetry index) of each dog before and after the treatment, as well as the difference between them. The last row shows the mean values of the group.

Dog ID	Pathology	SI before	SI after	Difference
1	DP + FH + FV	10.72	0.95	9.77
2	DP + FS	8.09	4.68	3.41
3	TV	12.12	4.17	7.95
4	DP	7.34	4.72	2.62
5	DP + FH + TV	10.34	5.84	4.50
6	DP + FS	8.70	4.32	4.37
7	DP + FS	9.93	0.57	9.36
8	DP	9.54	7.53	2.01
9	TV	9.56	3.40	6.16
10	DP + FH + TV	12.81	2.34	10.47
11	DP	9.32	7.02	2.30
12	DP + FH + TV	11.45	2.63	8.83
13	DP	9.47	8.39	1.08
14	DP + SF	7.57	4.55	3.03
Mean		9.78	4.36	5.42

DP: disc protrusion; DP + FS: disc protrusion + foraminal stenosis; DP + FH + TV: disk protrusion + flaval ligament hypertrophy + transitional vertebra; TV: transitional vertebra.

## Data Availability

The data presented in this study are available on request from the corresponding author. The data are not publicly available due to dog owners’ indications.

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
