# Peer review of "Platelet-Rich Plasma for the Treatment of Degenerative Lumbosacral Stenosis: A Study with Retired Working Dogs"

_animals, 2021, doi:10.3390/ani11102965_

Round 1
Reviewer 1 Report
I found the paper very clear and well structured. Although the use of PRP is becoming more widespread, the study proposes an alternative use of it.
I have some minor concern that I hope the authors can clarify, especially in dogs enrollment.
Line 74: the acronymous PRP has never been used in the introduction, please write it down in full
Line 148: even if antimicrobial activities of PRP has been demonstrated, do you perform a microbiological exam on PRP before injection? Do you completely exclude the possibility of a microbiological contamination?
Line 169: what kind of work or rehabilitation program was scheduled for dog after treatment? If a rest program was not used, please explain why.
Line 180: I know that in the study was included a low number of cases, but I think it could be interesting dividing dogs in groups by pathology (i.e. disk protrusion, transitional vertebrae etc.) and providing an evaluation of the effectiveness of treatment for each group. Is it possible to add a table?
Line 234: You stated that clinical signs were more severe in previous studies that evaluated the effectiveness of surgical decompression. Do you exclude from your study dogs whit severe clinical sings? Do dogs with severe paresis undergo to surgery? Please explain
Reviewer 2 Report
Comments to the Authors of manuscript number: animals-1408792 entitled “Platelet-Rich Plasma for the Treatment of Degenerative Lumbosacral Stenosis: a study with retired working dogs”.
The authors have presented very interesting data related to degenerative lumbosacral stenosis and its treatment with platelet-rich plasma in working dogs. It is very interesting description, which is worth to be published. I recommend this manuscript for publication after small correction. The manuscript is written very well and fits very well to chosen Journal-Animals.
- L 20- “this pathology” should be replaced by proper term.
- L 24 – “Animals” is dedicated only for the animals. This sentence should be omitted
- L 70- use the abbreviation
- L 74- the abbreviation used for the first time should be explained. Introduction is separate part from abstract
- L 125-127 this information should be given at the beginning of Materials and Methods
- The age also should be given., and how long they safer
- L 170-172 it should be replaced to the beginning of Materials and Methods
- L 175-179- the text is a bit chaotic. All basal information should be given at the beginning and in one the same place. Readers during reading should looking for all these information
- Is possible to present conventional radiography? It will be very interesting.
